# Evaluation of a Primary E-Health Intervention for People with Alcohol Use Disorder: Clinical Characteristics of Users and Efficacy

**DOI:** 10.3390/ijerph20156514

**Published:** 2023-08-03

**Authors:** Nathalie Stüben, Andreas Guenter Franke, Michael Soyka

**Affiliations:** 1Department of Psychiatry and Psychotherapy, University Hospital Munich, Ludwig Maximilian University of Munich, Nußbaumstr. 7, 80336 Munich, Germany; michael.soyka@med.uni-muenchen.de; 2University of Applied Labour Studies, Seckenheimer Landstr. 16, 68163 Mannheim, Germany; andreas.franke@arbeitsagentur.de

**Keywords:** alcohol, alcohol dependence, internet, web-based interventions, therapy

## Abstract

In Germany, only about 10% of patients with alcohol use disorder (AUD) are treated by the professional help system. “The First 30 Days without Alcohol”, an interactive e-health intervention, was developed to support people with “alcohol problems” to abstain from alcohol. The aim of this study was to examine the feasibility of the approach, the program’s target group, if and why it is effective. In March 2022 an email was sent to all users who had completed the program. A link to a web-based survey regarding the target group’s characteristics, its alcohol-use patterns, former attempts to change the problematic drinking behavior and experience with the program was introduced. The Alcohol Use Disorders Identification Test (AUDIT) was used prior and post intervention. A total of 718 participants completed the questionnaire. Of these, 99.2% suffered from AUD; 81.6% of participants were females, and about one third reported some form of psychiatric comorbidity; 46.6% did not use any additional help or assistance apart from the program; 78.3% reported to be abstinent after participation in the 30-day program, and the data show a significant AUDIT score reduction. Primary e-health interventions may contribute to the established addiction-help system. The intervention seems to reach predominantly highly educated and high-functioning females because of their characteristics.

## 1. Introduction

In Germany, according to the criteria of the Alcohol Use Disorder Identification Test (AUDIT), problematic alcohol consumption is present in 17.6% of the adult population, while 3.1% are alcohol dependent and another 2.8% have been diagnosed with alcohol abuse [1]. Substance use frequently leads to detrimental consequences such as somatic as well as mental disorders, accidental injuries, aggression, violence as well as suicide that could be avoided with abstinence [2]. Worldwide, 5.9% of deaths can be attributed to alcohol, e.g., cardiovascular disease, diabetes, injuries, gastrointestinal diseases or cancer [3]. Without using any therapeutic options, AUDs have a poor outcome [4,5]. Even if the health care system comprises a large number of services for people with alcohol-related disorders, there are severe deficits in addressing people with an AUD with the addiction-help system: only 10% of alcohol dependents can be reached with current therapeutic options, and the majority of people with AUD enter addiction therapy after many years of alcohol addiction [6]. The professional German addiction-help system is divided into free and open offers (e.g., addiction counseling, prevention or self-help) as well as care and treatment measures (e.g., withdrawal treatment, rehabilitation or integration assistance), which are assumed by costs and service providers (mostly public health or pension insurance). The majority of the facilities are run by the so called “Freie Wohlfahrtspflege” or other non-profit organizations. The other facilities are run by public or other organizations [7]. Even though people with AUD would benefit from early detection and intervention, the current German S3 guideline depicts that many addicts feel insecure and do not consider therapeutic options such as counseling and treatment at early stages of alcohol use [8]. As a consequence, a significant number of patients do not enter or enter too late the professional addiction-help system. There is a need for low-threshold and alternative therapeutic options with an emphasis on “picking up” people in an early state of “alcohol use” [8].

Web-based, electronic health interventions (e-interventions) are heterogenous with specific types of intervention and may play a crucial role in intervening in alcohol use and especially in AUD [9,10]. In recent years, web-based or e-interventions have been identified as promising new therapeutic tools in AUD [9,11]. E-interventions include cognitive-behavioral therapies [11], text messaging interventions [12,13,14] and web-based self-help programs [15], among others.

A novel approach is the establishment of abstinence-oriented interactive tools to address people with alcohol use problems with specific characteristics [16]. Regarding an analysis of components of web-based interventions, a recent systematic review demonstrated the following components to be relevant [10]: feedback and self-monitoring of behavior, instruction on how to adopt a behavior and social comparison. In light of this review, a recent Scandinavian study offering a cognitive behavioral online self-help concept demonstrated that privacy, anonymity and availability are important issues of online interventions [17]; moreover, the possibility of personal identification with others seems to be an important aspect as well [17].

Another study used asynchronized as well as synchronized chats with a counselor for a duration of ten weeks and found significant effects [18]. A recent meta-analysis demonstrated abstinence-promoting effects of web-based interventions in general [19]. Beyond that, web-based therapy can be considered as a stand-alone concept as well as in combination with conventional therapeutic options, which was supported by a meta-analysis about cognitive behavioral intervention with and without a combination of web-based interventions. The cognitive-behavioral criterion was met if the study intervention was described as CBT or Relapse Prevention, or included key elements of CBT, such as functional analysis, avoidance of high-risk situations and/or coping skills training [11].

“The First 30 Days without Alcohol” (30-day program) is one of the most frequently used e-interventions in the field of alcohol treatment among German-speaking people. It was developed in 2019 by Nathalie Stüben who was suffering from alcohol addiction, overcame it without using the professional addiction care system and started to publish web content about alcohol, AUD and abstinence. The online program consists of two parts: (1) cost-free information about addiction and abstinence via newsletters, Instagram, podcasts, YouTube and TikTok, and (2) the two paid online programs, “The First 30 Days without Alcohol” (30-day program) and the subsequent program “Stabilize Abstinence”. The programs support people with “drinking problems”—from risky drinking up to alcohol addiction—in living without alcohol. According to the primary goal of the current S3 guidelines [8], the 30-day program is completely abstinence-oriented. The programs consist of psychoeducation for disseminating information to increase participants’ knowledge as well as motivational work via an individual video approach, supporting materials and a guided online group for the digitalized personal interaction of users. The 30-day program offers tools to identify triggers and manage cravings, change dysfunctional thought patterns and identify the individual’s needs, as well as establish stamina and new routines in everyday life. Thus, the program is based on cognitive and behavioral psychotherapeutically effective techniques [20,21]. These include, for example, identification and prevention of high-risk situations (Relapse Prevention), methods for resource activation, drug refusal exercises and mindfulness-based strategies (mindfulness-based relaxation and imagery exercises) [22,23,24,25,26,27,28,29]. In addition, Nathalie Stüben, as a former sufferer, acts as a role model by authentically and credibly conveying and demonstrating that a life without alcohol is a gain and not a sacrifice.

The present study aims at examining the target group’s characteristics, its alcohol-use patterns, their former attempts to change the problematic drinking behavior and whether the program is effective as well as which parts of the program’s design make it effective. The characteristics of the target group included sociodemographic data and physical as well as psychological comorbid disorders. The latter is of interest because alcohol use disorder is associated with various mental disorders, such as affective disorders, anxiety disorders or attention deficit hyperactivity disorder (ADHD) [30,31]. It was of interest to examine whether this is also the case in this sample. The present study included a quantitative design. Descriptive statistics were used to describe the sample in terms of specific characteristics and frequencies, followed by the performance of significant analytic procedures to examine intraindividual differences in the paired sample. The study was conducted in accordance with the Declaration of Helsinki, and the protocol was approved by the Ethics Committee of the Ludwig Maximilian University of Munich (No. 23-0120 KB).

## 2. Methods

The 30-day program has been online since October 2019. On the 27 March 2022, a one-time email went out to everyone who had completed the program by then, introducing a link leading to a web-based survey questionnaire on the domain “oamn.jetzt”. The questionnaire offered multiple choice questions regarding the target group’s characteristics, alcohol-use patterns, former attempts to change the problematic drinking behavior, and experience with the program. In addition, the Alcohol Use Disorders Identification Test (AUDIT) was used prior and post intervention. The survey was closed two weeks later. 

### 2.1. Data Acquisition

Data were collected using the tool “Typeform”. Prior to participation, participants were informed about aims, contents and background of the study as well as about anonymous data handling and the voluntary character of participation.

Initially, independent data were collected (sex, age, highest degree of education). Then, participants were asked about smoking behavior, physician’s diagnosis of any psychiatric disorder and whether abstinence reduced symptoms of the above-mentioned disorder. After that, the Alcohol Use Disorders Identification Test (AUDIT) as a WHO-recommended tool to assess severity of alcohol use [24] was used; its questions were asked regarding (a) before completing the 30-day program and (b) after having completed this program. Thereafter, the participants were asked non-standardized questions about their source of knowledge of the 30-day program, reasons for choosing it, learning effects of the program, consequent implementation of its tasks, recognized changes during the program, feelings while quitting to drink, completion of the program, date of completion and (re-)lapses during the program. Furthermore, participants were asked about use of other interventions for abstinence, subjective feeling of mental stability as well as likelihood of recommendation of the program to others. At the end participants were asked to click on “submit” to confirm voluntary participation.

### 2.2. Data Analysis

SPSS Statistics Version 28.0.1.0 was used for statistical analysis.

## 3. Results

The call for participation was sent to all users of the 30-day program (*n* = 2943). The questionnaire was accessed by 1515 participants (51.5%). The questionnaire was started by *n* = 1019 participants (34.6%) and submitted by *n* = 746 (25.3%) participants. *N* = 718 agreed to data use for scientific evaluation leading to a completion rate of 24.4% with an average completion time of 15.47 min.

### 3.1. Characteristics of The Target Group

Participants were 49 years old (mean, SD = 9,7); the youngest was 23, the oldest 78. The vast majority were female (male: 18.4%, *n* = 132; female: 81.6%, *n* = 586) and highly educated (university degree 47.8%, *n* = 343; baccalaureate/high school degree 25.9%, *n* = 186; secondary school degree: 22.0%, *n* = 158; secondary/main school: 4.0%, *n* = 29; no school degree: 0.3%, *n* = 2). In sum, 32.2% (*n* = 174) were smokers, 20.8% (*n* = 149) had never smoked, 55.0% (*n* = 395) quit in the past. 

The majority (67.8%, *n* = 487) had not been diagnosed with any psychiatric disorder (except substance use); 32.2% (*n* = 231) suffered from any psychiatric disorder. Regarding those who stated that they have any psychiatric disorder, 52.8% (*n* = 180) reported depression, 22.9% (*n* = 78) anxiety disorder(s), 12.9% (*n* = 44) eating disorders, 8.8% (*n* = 30) post-traumatic stress disorder (PTSD) and 2.6% (*n* = 9) attention deficit/hyperactivity disorder (ADHD).

Regarding remission of symptoms of the above-mentioned psychiatric disorders during abstinence, participants indicated that 10.4% (*n* = 24) had not been (completely) abstinent; 50.2% (*n* = 116) stated that symptoms were significantly reduced; 10.8% (*n* = 25) reported that all symptoms disappeared; 16.5% (*n* = 38) reported that symptoms were unchanged; and 11.3% (*n* = 26) could not decide.

When participating in the survey, the vast majority indicated having quit drinking completely (78.3%, *n* = 562), while 29.7% (*n* = 213) were abstinent for 0–3 months, 21.3% (*n* = 153) for 4–6 months, 13% (*n* = 93) for 7–12 months and 14.3% (*n* = 103) for over 12 months. However, 21.7% (*n* = 156) had not quit yet.

Moreover, 30.9% (*n* = 222) started the program during the last 3 months, 44.4% (*n* = 319) before more than 3 months ago but less than 1 year and 24.7% (*n* = 177) before more than 1 year.

### 3.2. Evaluation of The Program

According to Babor et al., the AUDIT [24] total score cutoffs indicate different AUD severity levels. A score of 0–7 refers to low-risk drinking or abstinence, 8–15 indicates risky use, a score of 16–19 corresponds to harmful use and a score of 20 or more indicates probable dependency. Table 1 shows the before and after results according to this cluster.

Regarding the mean outcome values, the total AUDIT score was 22.6 (SD = 6.56) before participating in the 30-day program. After having completed the program, the AUDIT score was significantly reduced to a mean of 3.26 (SD = 6.19) (*p* < 0.01).

The most significant results were the reduction of the frequency of drinking (means before and after: 3.58 vs. 0.64), the amount of drinks (2.12 vs. 0.29) and the frequency of drinking more than six drinks (2.80 vs. 0.39). Table 2 shows these results translated to the categories “improved”, “unchanged” and “worsened”, making them comparable to data of the current Statistical Report on Substance Abuse Treatment in Germany [26].

The majority of participants knew about the existence of the 30-day program via YouTube (18.9%, *n* = 136), TV (18.0%, *n* = 129), print and online articles (17%, *n* = 122), search engines such as Google (17.0%, *n* = 122) and the above-mentioned podcast (14.5%, *n* = 104). Other types of media (radio: 1.3%, *n* = 9; book: 12.1%, *n* = 87) and recommendations (4.9%, *n* = 9) were rather infrequent ways of becoming aware of the 30-day program.

There were multiple reasons for deciding to participate in the 30-day program such as a positive attitude towards future abstinence (24.1%, *n* = 503), the possibility to stay at home (not to go out) (19.9%, *n* = 416) or to integrate the 30-day program with everyday life (18.5%, *n* = 385), a feeling of “mismatch” between oneself and the professional addiction help system (15.8%, *n* = 330), the opportunity to start “right now” (13.5%, *n* = 281), increased awareness around having an alcohol problem (6.9%, *n* = 143) and a personal recommendation of the program (1.3%, *n* = 28).

Out of all participants, 44.6% (*n* = 320) “enjoyed” the program, 36.1% (*n* = 259) stated that they “really enjoyed” the program; others could not decide or did not enjoy the program.

Learning effects were stated to be high in general: Only 0.4% (*n* = 10) stated to have learned nothing. All other participants stated to have learned that abstinence can be considered as a kind of benefit (20.5%, *n* = 562), to be not alone with the alcohol problem (7.9%, *n* = 490), the important role of alcohol in one’s own life (16.9%, *n* = 464), to change their way of thinking (16.0%, *n* = 440), a new strategy for abstinence (14.9%, *n* = 409) and to understand oneself in a better way (13.5%, *n* = 370).

Specific components of the 30-day program were considered to be helpful. These results are shown in Table 3.

Most of the participants had completed all (25.5%, *n* = 183) or almost all (38.2%, *n* = 274) of the required tasks; 13.5% (*n* = 97) only watched the program’s videos and read the respective texts and only 1.1% (*n* = 8) did not use the provided material.

Nearly all participants recognized “pronounced positive changes” (47.2%, *n* = 339) or “positive changes” (45.0%, *n* = 323), and only 6.3% (*n* = 41) recognized no or nearly no or even negative changes. For specific changes in their different aspects of life, see Table 4.

Using the 30-day program, it was “easier” (37.0%, *n* = 266) or “significantly easier” (26.5%, *n* = 190) than expected to quit drinking for the majority of all participants. However, 16.3% (*n* = 117), respectively, had not quit completely when answering the questions.

For alcohol-dependent subjects, the question of lapses and relapses is a highly important aspect that was assessed in the survey, too. Of the subjects, 23.4% (*n* = 168) experienced lapses or relapses after having started the program, as presented in Table 5.

To abstain from drinking, nearly half of all participants (46.6%, *n* = 402) did not use any additional help or assistance. Some used the second program “Stabilize Abstinence” (23.9%, *n* = 206) or sought additional help from a psychotherapist (11.6%, *n* = 100), a physician (3.9%, *n* = 34), an addiction counseling center (5.8%, *n* = 50), a self-help group (4.9%, *n* = 42), other online offers/programs (3.29%, *n* = 28).

A significant part of participants stated that they would recommend the 30-day program to others (Likert scale: very likely/probably: 72.8%, *n* = 523; likely/probably: 15.5%, *n* = 111, indifferent: 5.6%, *n* = 40; unlikely/improbable: 1.5%, *n* = 11; very unlikely/improbable: 4.6%, *n* = 33).

## 4. Discussion

This study examines clinical characteristics and outcomes of alcoholic patients using a “pure” online digital intervention without prior personal or medical screening. Key findings of this study:In a short two-week run-in period, 1019 (34.6%) of 2943 participants started the detailed questionnaire, about half of the users assessed the questionnaire and 718 (24.4%) submitted data indicating a major interest of participants in providing relevant data.Of these, 99.2% suffered from AUD as indicated by an AUDIT score of 8 or higher. The mean AUDIT score of 22.61 (SD = 6.6) when entering the program was significantly reduced to a mean of 3.26 (SD = 6.19) (*p* < 0.01) after having finished the program.Contrary to other studies, 81.6% of participants were females indicating a strong need from female AUD patients for digital online interventions.About a third of patients reported some form of psychiatric comorbidity.About 46.6% did not use any additional help or assistance apart from the program “The First 30 Days without Alcohol”, while 23.9% used the follow-up program “Stabilize Abstinence”.About 78.3% reported to be abstinent after participation in the 30-day program.

The study shows that users of the present digital intervention have a high media affinity and are predominantly female and well educated. Regarding their characteristics, they differ significantly from the “standard” German patient in the field of AUD. The Statistical Report on Substance Abuse Treatment in Germany [32] provides an overview of the current situation regarding substance abuse, consisting of outpatient and inpatient data of German patients. The comparison of the core results on the “standard“ German patient in the field of AUD shows the difference to the 30-day program participants. While the age (mean) is about the same (30-day program: 49; outpatient treatment: 46, inpatient treatment: 47), the main differences occur in the amount of women attending the program (30-day program: 81.6%; outpatient treatment: 2%; inpatient treatment: 31%) and in the educational attainment levels, having at least baccalaureate/high school degree (30-day program: 73.7%; outpatient treatment: 18.8%; inpatient treatment: 19.6%).

Regarding psychiatric comorbidities in patients entering treatment, there is a difference as well (30-day program: 32.2%; outpatient treatment: 52.7%; inpatient treatment: 80.5%). The majority of those who quit alcohol use completely with the help of the 30-day program appreciated the positive way of thinking about abstinence followed by the decentralized possibilities to use the 30-day program as well as the personal communication style combined with learning tools about alcohol and the possibilities of interaction. The majority (really) enjoyed using the program and evaluated it to be a real profit for them. However, (re-)lapses occurred during the 30-day program as well as after having finished the program. Another significant difference relates to the number of drinks consumed before and after the program respectively treatment, which mostly improved (30-day program: 88.3%; outpatient treatment: 49.7%; inpatient treatment: 69%). Few were unchanged (30-day program: 11.2%, outpatient treatment: 45.2%; inpatient treatment: 29.5%), or worsened (30-day program: 0.4%, outpatient treatment: 5.1%; inpatient treatment: 1.5%).

For decades, therapeutic options for alcohol dependence have been dominated by self-help strategies. Meanwhile, there is a plethora of studies, reviews and meta-analyses about therapeutics [33,34,35]. Although interventions in primary care are crucial in diagnosis and treatment of AUD, few treatment approaches were demonstrated to be successful. For example, pharmacological treatment strategies are of subordinate importance [3]; a recent network meta-analysis revealed that only acamprosate is effective in primary care [36,37]. In various meta-analyses, motivational interviewing, cognitive and behavioral therapy, contingency management, family as well as marital therapy and brief interventions were found to be significantly effective [8], among others. The main aim of all interventions is abstinence from alcohol; however, reduced drinking as a harm reduction approach is widely accepted as a secondary aim [8].

In spite of the broad range of therapeutic options, only about 10% of alcohol dependent patients can be reached by these options [38] and the majority of people with AUD enter addiction therapy after many years of alcohol use [6]. The reasons for reaching only 10% of AUD patients could be underlined at least in part by the present study: 16% stated that they do not feel like they are fitting into the professional addiction help system. Web-based interventions represent a novel approach for identification, motivation and treatment of patients with AUD—at least in supporting existing treatment options. They are available around the clock, from everywhere around the world, have the highest degree of privacy and allow autonomous decision about whether to use it and the time and frequency of using it, the latter being a crucial selection criterion [39]. This was confirmed by the present study depicting that the most important reasons to decide on the 30-day program were the possibility to stay at home, to integrate it into one’s daily life and the opportunity to start “right now”. Furthermore, the opportunity to use web-based interventions in parallel to any other intervention was confirmed by the present study: 47% did not use any other intervention, but others did. Overall, there is a paucity of systematic data about the use of different treatment offers. Moreover, participants stated that they favor the personal approach of the 30-day program, the background knowledge, the tasks and the interaction (contact possibilities to the program’s background team, continuous contact to the moderated online group).

In contrast to the easy way of providing (medical) information with web-based interventions, interaction is more difficult but crucial regarding effectiveness [17]. Nevertheless, the 30-day program contains, at least in part, interactive elements. For example, participants are encouraged to take part in the moderated online group, e.g., via posting about their reason to quit drinking, their experience with difficult situations they have overcome or yet to face.

Today, there are many more web-based interventions than in the past; these interventions are increasingly analyzed and evaluated [40,41,42,43,44]. They are useful in different stages of alcohol use and in different ways: early detection, early intervention, stand-alone therapy, complementing other therapeutic options in different stages, supporting motivation or follow-up care. However, web-based interventions promote interaction among their users. This may strengthen their self-efficacy expectation.

The main characteristics of web-based interventions are that it is a low threshold to use these interventions, they offer permanent availability and they provide the possibility to adopt an active role in one’s own (pathologic) pattern of alcohol use. These aspects may have strengthened digital interventions during the COVID-19 pandemic and its lockdowns due to the unavailability of any (local) services. Beyond that, most recent web-based interventions such as the 30-day program enable interaction with others and feedback. However, there may remain a paucity of “real” therapeutic empathy. Nevertheless, web-based interventions in the field of AUD apparently have a promising future. This future will probably not comprise a “pure” digitalization of existing therapeutic interventions but developing, new types of interventions and materials. Of course, there is an epidemic need and a broad interest.

There are some limitations of this study to be discussed. This study has to be viewed as a proof-of-concept study integrating 718 participants out of 2943 users of the 30-day program. As in other anonymous web-based studies, generalizability of findings can be questioned. Furthermore, web-based surveys do not allow control of the sample regarding participation (participation bias) which must be considered when interpreting the results; this is valid for personal characteristics of participants as well as their attitudes, their success of abstinence, tendency to minimize the problem, tendency of socially desirable answers and further aspects. However, it can be assumed that all users of the 30-day program identified a problem with their drinking pattern. This may lead to more reliable answers concerning alcohol consumption. Beyond that, the high degree of privacy may strengthen participants’ honesty. Another limitation of the study is that the program is not accessible to everyone, as one has to pay for the program. This limits the representativeness of the sample of people with AUD by excluding those who do not have the financial resources. In addition, it is believed that people who spend money on such a program are more likely to be among the motivated. Motivation is a key factor in successful abstinence and may be a prerequisite for program effectiveness. At the same time, the financial investment may have increased the motivation to implement the program.

Overall, these preliminary data indicate that many people who were not attracted by the professional addiction help system may profit from digital online interventions such as the 30-day program. However, the long-term effects of such interventions are not yet clear. Therefore, long-term studies including control groups are necessary to further explore the benefits of digital online interventions.

## 5. Conclusions

In conclusion, the present study shows that the interactive web-based program “The First 30 Days Without Alcohol” has a positive effect on reducing AUD symptoms. Most participants were abstinent after the program. Specifically, women appeared to be attracted to the interventions presented. Due to the low-threshold and anonymous offer, people with AUD can be reached and supported who have not yet arrived in the established addiction support system. The results give the first and many promising indications of the effectiveness of the program. Based on this, randomized controlled trials should be conducted to examine these initial efficacy study results.

## Figures and Tables

**Table 1 ijerph-20-06514-t001:** Severity levels of AUD indicated by AUDIT score.

Severity	Before Program	After Program
Low risk or abstinence	*n* = 6 (0.8%)	*n* = 598 (83.3%)
Risky	*n* = 101 (14%)	*n* = 75 (10.4%)
Harmful use	*n* = 114 (15.9%)	*n* = 17 (2.4%)
Dependence	*n* = 497 (69.2%)	*n* = 28 (3.9%)

**Table 2 ijerph-20-06514-t002:** Drinking quantity before and after completing the 30-day program.

Alcohol Consumption	Improved	Unchanged	Worsened/Increased
Frequency of drinking	*n* = 664 (92.5%)	*n* = 54 (7.5%)	*n* = 0 (0%)
Number of drinks	*n* = 634 (88.3%)	*n* = 81 (11.2%)	*n* = 3 (0.4%)
Frequency of drinking more than 6 drinks	*n* = 629 (87.6%)	*n* = 88 (12.3%)	*n* = 1 (0.1%)

**Table 3 ijerph-20-06514-t003:** Degree of usefulness of the program’s components.

	Number (*n*)	Percentage (%)
Personal approach	502	16.0%
Background knowledge	505	16.1%
Tasks to integrate in everyday life	364	11.6%
Daily routine	400	12.8%
Diary	308	9.8%
Audio training (motivational talks, affirmations, etc.)	282	9.0%
Program‘s moderated online group	300	9.6%
Contact to the program’s background team	100	3.2%
Continuous access to the program’s content	367	11.7%

**Table 4 ijerph-20-06514-t004:** Prevalence of aspects of life improved.

Situation	Number (*n*)	Percentage (%)
Stress	10.1%	472
Depressive mood	8.4%	393
Self-confidence	9.4%	438
Confidence about future	8.7%	406
Relationship to others	7.8%	366
Mental skills/capabilities	8.7%	409
Fitness and condition	6.4%	301
Professional capabilities	6.8%	319
Dealing with difficult emotions	7.5%	353
Sleep and sleep quality	10.3%	481
Somatic complaints	5.2%	242

**Table 5 ijerph-20-06514-t005:** Lapses and relapses.

	Lapses		Relapses	
	Number (*n*)	Percentage (%)	Number (*n*)	Percentage (%)
During the program	50	7%	62	8.6%
During first month after finishing the program	39	5.4%	35	4.9%
During the first 3 months after finishing the program	63	8.8%	32	4.5%
During the first 6 months after finishing the program	22	3.1%	18	2.5%
During the first 12 months after finishing the program	12	1.7%	11	1.5%
After more than 12 months after finishing the program	11	1.5%	2	0.3%

## Data Availability

The data presented in this study are available on request from the corresponding author. The data are not publicly available due to privacy restrictions.

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
