# Peer review of "Evaluation of a Primary E-Health Intervention for People with Alcohol Use Disorder: Clinical Characteristics of Users and Efficacy"

_ijerph, 2023, doi:10.3390/ijerph20156514_

Round 1
Reviewer 1 Report
The authors of “Evaluation of a primary e-health intervention for people with alcohol use disorder: Clinical characteristics of users and efficacy” examine an e-health intervention for alcohol use. Authors describe the users of this intervention as well as why the users find it to be helpful. Examination of e-health interventions are important and timely. However, the authors could improve the manuscript by making some edits and answering some questions. Generally, the manuscript would benefit from a thorough proofreading and correction of grammatical errors.
Introduction
The authors do not include a citation when reporting the number of individuals who may report binge drinking over the last month. They also don’t cite that binge drinking involves consuming that number of drinks over a period of 2-3 hours. Both of these items should be remedied.
For readers not based in Germany, the manuscript would benefit from a short description of the addiction help system.
The introduction would benefit from the authors including more information regarding the specific e-interventions that were found to be beneficial in the cited meta analysis.
The manuscript would benefit from consistent report of aims across abstract and introduction and then clear analyses that address aims. For example, there is no mention of dual diagnosis in the introduction or abstract.
Methods
It’s unclear what other portions of the questionnaire use standardized measures aside from the AUDIT. How were the questions and responses arrived at?
Results
The results could benefit from organization that mirrors discussion of aims in the abstract and introduction.
A reported completion rate of 70.5% seems misleading when 2,943 surveys were sent out and 746 were submitted in full.
“The majority of all participants felt more or rather less…using the latter. This is a sentence I found confusing.
Discussion
Limit repetition in the numbered points.
It may be helpful to state that the “standard” patient being described is a “standard” German patient.
One limitation that isn’t discussed is the fact that users must pay to use the program. I wonder how this makes for a different sample than if the program was provided free of cost.
Authors conclude that “many people who cannot be addressed by the professional addiction help system…” To my knowledge this isn’t a sample of individuals that sought help through the professional addiction help system and so I believe this statement should be adjusted and conclusions tempered.
As mentioned in comments and suggestions, the manuscript should be thoroughly reviewed for grammar.
Reviewer 2 Report
The paper assess the effectivity of an interactive e-health intervention to trait alcohol use. The paper is clear and the data are well presented.
I have only two concerns:
One about the description of the e-health intervention. It is not so clear what therapeutical principles are based?, How works?
The other concern is about discussion, I found it so large and I think that conceptualization aobut effectivity of e-health as compared to other treatments should be less large.
